# Exploring Treatment for Depression in Parkinson’s Patients: A Cross-Sectional Analysis

**DOI:** 10.3390/ijerph18168596

**Published:** 2021-08-14

**Authors:** Elisabeth C. DeMarco, Noor Al-Hammadi, Leslie Hinyard

**Affiliations:** 1Department of Health & Clinical Outcomes Research, Saint Louis University School of Medicine, St. Louis, MO 63104, USA; elisabeth.demarco@health.slu.edu (E.C.D.); noor.alhammadi@health.slu.edu (N.A.-H.); 2Advanced HEAlth Data (AHEAD) Institute, Saint Louis University, St. Louis, MO 63104, USA

**Keywords:** Parkinson’s disease, depression, National Health and Nutrition Examination Survey, NHANES, Centers for Disease Control and Prevention, CDC

## Abstract

Depression is a highly prevalent, often underrecognized and undertreated comorbidity of Parkinson’s disease closely correlated to health-related quality of life. National trends in depression care for patients with Parkinson’s disease are not well documented. This paper identifies a cohort of patients with Parkinson’s disease from nationally representative survey data and analyzes trends in depression care. Using data from the 2005–2006 through 2015–2016 waves of the National Health and Nutrition Examination Survey (NHANES), individuals were classified as Parkinson’s patients by reported medication use. PHQ-9 scores were used to identify individuals screening positive for depression. A composite treatment variable examined the reported use of mental health services and antidepressant medication. Survey participants with probable PD screened positive for depression, reported the use of antidepressant medication, and reported visits to mental health services more frequently than the control group. Survey participants with PD who screened positive for depression were more likely to report limitations in physical functioning due to an emotional problem than controls. While depression is highly prevalent among individuals with Parkinson’s disease, they are more likely to receive any treatment. Further research is required to investigate differences in patterns of treatment, contributing factors of emotions to limitations in physical functioning, and appropriate interventions.

## 1. Introduction

Parkinson’s disease (PD) is a chronic neurodegenerative disease with characteristic motor and non-motor symptoms affecting 572 per 100,000 people aged 45 or older and projected to impact 1,238,000 U.S. individuals by 2030 [1]. Depression is a highly prevalent comorbidity of PD, with prevalence estimates ranging from 22% to 91% [2,3,4] and estimates of concomitant anti-parkinsonian and antidepressant use in older adults ranging from 25% to 58% of patients [5,6], with selective serotonin reuptake inhibitors emerging as the most frequently used antidepressant [5,6]. Moreover, depression is a key predictor of health-related quality of life, with increasing severity corresponding to decreased quality of life [7,8]. While depression principally impacts mental and emotional aspects of quality of life [9], it may also exacerbate motor symptoms [10] and alter patient self-perception of motor function [11]. Depression severity is more strongly influenced by patient perception of disability than neurologist-assessed motor function [12], establishing the potential for a cycle of increased perceived disability and increasingly severe depression, which may ultimately drastically decrease quality of life.

Despite the impact on patient life, depression in PD patients is often underdiagnosed [13,14,15], in part due to overlapping somatic symptoms [10,16,17] and lack of systematic screening [18], and may contribute to undertreatment of this condition [15]. In a German study, 75% of patients suffering from moderate to severe depression were not prescribed antidepressants [18]. A similar study of patients treated at the National Parkinson’s Foundation Centers of Excellence found 54% of patients meeting depression criteria received treatment, defined as either antidepressant use or counseling by a mental health professional or social worker [19]. Those patients who were treated reported improvements in mobility and execution of activities of daily living [11] and improved quality of life [20], despite no change in clinician-assessed capabilities [11] or PD disease severity [20]. Irrespective of actual gains in motor function, treating depression has great potential to improve patient quality of life.

There is no nationally representative assessment of the prevalence of depression among patients with Parkinson’s disease. Thus, it remains difficult to assess the prevalence of treatment (or undertreatment) of depression in this population. Although current literature indicates a continued unmet need for treatment among depressed Parkinson’s patients, few studies address the use of antidepressants in younger patients prescribed anti-parkinsonian medication or directly compare depression treatment in Parkinson’s patients with the general population. Additionally, depression treatment in recent studies of Parkinson’s patients has been defined primarily by pharmacologic treatment [6], although psychotherapeutic approaches are also available for depression treatment [7,21] and endorsed by a physician focus group [22].

We identified a cohort of Parkinson’s patients from a nationally representative survey sample to determine depression prevalence and trends in depression care. We then compared both pharmacologic and nonpharmacologic depression care in adults with Parkinson’s to depression care in non-Parkinsonian adults.

## 2. Materials and Methods

### 2.1. Data Source

Data from the National Health and Nutrition Examination Survey (NHANES) years 2005–2016 were used for this analysis [23]. Sponsored by the National Center for Health Statistics, a division of the Centers for Disease Control, NHANES is designed to assess both the health and nutritional status of children and adults in the United States. Groups including African Americans, Hispanics, and those over the age of 60 were oversampled to generate reliable data. Data collection occured in two stages—an interview conducted within the participant’s home and an examination at a mobile study center. Interview questions covered demographics, socioeconomic markers, dietary habits, and health-related questions. The examination included laboratory tests and physiological, medical, and dental measurements.

### 2.2. Study Design and Participant Sample

This analysis drew on data from the past 10 years of available data: waves 2005–2006 through 2015–2016. All participants 20 years of age or older at the time of the survey who completed both the household questionnaire and Mobile Examination were included in the study. PHQ-9 was administered to participants 12 years of age and older at the Mobile Examination Center. All other data were collected during the household interview. Participants who completed only the household interview component of NHANES and those with missing information on the outcome or predictor variables of interest were excluded (*n* = 14,189; Figure 1). Participants completed an informed consent at the time of survey, and the data are publicly available. The current study was deemed exempt by the Saint Louis University Institutional Review Board.

### 2.3. Measures

The primary outcomes for this study were pharmacologic therapy with conventional antidepressants, self-report of discussing mental health with a provider in the last 12 months [24], and limitations in physical functioning due to depression, anxiety, or another emotional problem (Figure 1). Depression management was described as a composite treatment variable, where participants were considered to have some sort of depression management if they were deemed to be receiving pharmacologic treatment or reported discussing mental health with a provider in the last 12 months.

Pharmacologic treatment was determined using the “Prescription Medication” module, where participants were asked “Have you taken or used any prescription medication in the last month [excluding dietary supplements]?” Participants could answer “yes”, “no”, “I don’t know”, or decline to answer. Those who responded “I don’t know” or declined to answer were excluded from this analysis. Participants who indicated use of a prescription medication were then asked for additional details of any medications taken, including the number of medications, the length of time using the medication, and the generic drug name. All generic drug names were subsequently identified with a specific NHANES code. A list of antidepressant medications (Appendix A) was created from the codes recorded by NHANES. Any individual who reported they had taken one of the included drugs in the last month was classified as receiving pharmacologic treatment for depression.

Self-report of an encounter with a mental health professional was recorded in the “Hospital Utilization and Access to Care” module. Participants were asked “During the past 12 months, that is since [the current month] of [the last year], have you seen or talked to a mental health professional such as a psychologist, psychiatrist, psychiatric nurse or clinical social worker about your health?” Participants could respond “yes,” “no,” “I don’t know,” or decline to answer. Those who responded “I don’t know” or declined to answer were excluded from this analysis. Those who responded “yes” were considered to have experienced mental health treatment in the past year.

Limitations in physical functioning due to depression, anxiety, or other emotional problems were recorded in the “Physical Functioning” questionnaire module. Participants were asked, “Are you limited in any way in any activity because of a physical, mental, or emotional problem?” Participants could respond “yes”, “no”, “I don’t know”, or decline to answer. A follow-up question—“What condition or health problem causes you to have difficulty with or need help with [activities]?”—was asked to determine the causative condition. Those who reported difficulty caused by “depression, anxiety, or an emotional problem” were considered to have limitations in physical functioning related to their mental health. Any response outside of “depression, anxiety, or other emotional problem” indicated a decline in physical functioning unrelated to mental health. Those who responded “I don’t know” or declined to answer were excluded from this analysis.

Parkinson’s disease and positive depression screen at the time of data collection were considered independent variables of interest. Participants were classified as having Parkinson’s disease if they reported the use of one or more medications indicated for the treatment of PD (Appendix A) [24]. This determination was made using the responses to questions concerning prescription medication, as discussed in pharmacologic treatment of depression. This identification method was limited by the medications NHANES records and codes and required individuals to be actively treated for PD to be classified as having PD. Any participant who did not report taking an identified anti-parkinsonian medication was classified as not having PD.

All participants were screened for symptoms of depression as part of Mobile Examination Center interview data collection using the Patient Health Questionnaire-9 (PHQ-9; [25]). It was used here to delineate those participants who screened positive for depression at the time of survey. PHQ-9 scores of 5, 10, 15, and 20 represented mild, moderate, moderately severe, and severe depression, respectively [25,26]. At the time of the survey, participants responding affirmatively to having suicidal ideation (“Over the last 2 weeks, how often have you been bothered by the following problems: Thoughts that you would be better off dead or of hurting yourself in some way?”) or displaying signs of emotional distress upon viewing this question were assessed by the Mobile Examination Center staff physician and referred for mental health services as needed [27].

### 2.4. Statistical Analysis

All analyses were conducted in SAS 9.4 (SAS Institute Inc., Cary, NC, USA) using the survey method provided by the National Center for Health Statistics (NCHS) to account for the complex sampling design of the five snapshots included for this analysis. The distribution of baseline characteristics was reported as weighted percentages and comparisons were determined using the chi-square analysis with the SURVEYFREQ procedure. Weighted estimates were generated by the analysis to represent the prevalence and trends of disease in the U.S. population. Odds ratios were calculated using a logistic regression model.

Dichotomous variables were created for the patients who received medications for Parkinson’s disease, received medications for depression, reported symptoms of depression, sought professional help for depression, and had physical impairment related to depression. For the reported symptoms of depression, a cumulative PHQ-9 score was calculated, and a cutoff of 10 or greater was used to identify an individual with depression for this analysis [25,26]. Covariates such as age, annual household income, poverty level, and insurance status were also categorized, as appropriate.

## 3. Results

A total of 18,731 participants met our inclusion criteria. In our sample, participants ranged in age from 20 years to over 80 years of age, with the majority between ages 40 and 59 (Table 1). In total, 57.7% of participants were female, and non-Hispanic white individuals predominated compared to other ethnicities (75.34%; Table 1). Economic status was determined using the ratio of family income to the federal poverty line for the given year. While 56% of participants earned less than $45,000 annually, 87.73% of participants were determined to be above the poverty line, as family income was greater than the poverty line. A total of 95% of survey respondents were U.S. citizens and 63.92% were married or living with a partner. In total, 54% of respondents had graduated from high school, while an additional 30% had completed a college degree.

A total of 418 participants reported use of anti-parkinsonian medications. These individuals were statistically significantly different from those not reporting the use of an anti-parkinsonian medication across all demographic variables examined. Individuals reporting the use of anti-parkinsonian medications were older, more often female (65% in the PD group vs. 57% in the non-PD group), and a greater proportion had not achieved a high school degree (Table 2). In total, 58.8% of respondents taking an anti-parkinsonian medication had an annual income greater than or equal to $45,000, compared to 43% of those not taking such medication (Table 2). While 87.92% of non-Parkinsonian individuals reported a family income greater than the poverty line, a lesser proportion (79%) of those taking anti-parkinsonian medication reported a family income exceeding the poverty line (Table 2). A total of 26.46% of those classified as Parkinson’s patients screened positive for depression on the day of the exam, compared to 9.58% of those not classified as PD patients (Table 2). 

In total, 1960 participants screened positive for depression (PHQ-9 score ≥ 10). Of those participants with a PHQ-9 score ≥ 10, 5.75% reported use of an anti-parkinsonian medication (Table 3). The odds ratio (OR) for screening positive for depression among those taking anti-parkinsonian medication is 3.309 (95% Confidence Interval: 2.630–4.162) compared to those not taking anti-parkinsonian medication (Figure 2).

Overall, those reporting the use of anti-parkinsonian medication more frequently reported management of depression by medication or medical encounter with a mental health provider compared to those without PD (85.12% vs. 21.5%, respectively; Table 3). Those classified as Parkinson’s patients who screened positive for depression more frequently reported use of antidepressants (62.42% vs. 21.49% of controls; Table 3). Additionally, there were statistically significant differences in reports of consultation with mental health providers and limitations in physical function as a result of mental or emotional health between those using anti-parkinsonian medication and those who did not report the use of this medication. Among those screening positive for depression, those reporting the use of an anti-parkinsonian medication were also associated with higher odds to report depression management with either medication or medical encounter (OR 4.478; 95% CI: 2.757–7.274; Figure 2) and report limitation in physical functioning due to depression, anxiety, or another emotional problem (OR 2.237; 95% CI: 1.294–3.866; Figure 2). 

## 4. Discussion

This study identified a nationally representative sample of individuals with Parkinson’s disease, about 2% of those surveyed by NHANES from the 2005–2006 wave to the 2015–2016 wave. The survey sample identified in this study affirmed that depression is a prevalent comorbidity of PD, with those classified as PD patients more frequently reporting the use of antidepressant medication and consultation with a mental health provider (Table 3, Figure 2).

Reports of prevalence of depression among PD patients vary greatly throughout the literature, depending on how depressive symptoms or disorders are classified and the study population. In this analysis, 26.46% of the survey sample screened positive for depression, a significant difference compared to 9.94% of those not classified as PD patients (Table 2), consistent with previously reported ranges [2,3,4,5,6]. The prevalence of depression among non-PD patients in our study is similar to the reported prevalence of depression in the general population (8.1%; [28]). Those participants taking anti-parkinsonian medication were about threefold more likely to screen positive for depression (Figure 2), indicating depression remains a significant burden for those diagnosed with PD and greater than in the general population.

Underdiagnosis, underrecognition, and undertreatment of depression within this population continue to be reported [10,18,19]. Among those PD patients screening positive for depression, a majority reported antidepressant use, and more than a third reported consultation with a mental health provider (Table 3). While 85.12% of those reporting use of anti-parkinsonian medication and screening positive for depression indicated some form of management for depression, 14.88% of these patients remain untreated (Table 3). This is markedly different from previous studies, where 40–75% of depressed individuals with PD reported no receipt of depression treatment [18,19]. Among U.S. adults with a lifetime diagnosis of major depressive disorder, approximately 30% do not receive any sort of depression management, pharmacologic or otherwise [29,30]. The World Health Organization defines a treatment gap as the proportion of individuals qualifying for treatment who do not receive it and concluded that 42% of North Americans with an affective disorder remained untreated [31]. In comparison to the general population, therefore, more PD patients are receiving some sort of depression management, and use of anti-parkinsonian medication is positively associated with the receipt of some sort of depression treatment (OR 4.48; Figure 2). Patients with PD more frequently attend neurology and primary care appointments [32] and have increased rates of health utilization compared to patients without PD [33]. Compared to the general population, patients with PD take approximately twice as many medications, and the differences between these groups in healthcare utilization and medication usage increase with increased severity of Parkinson’s disease [33]. As the current study relies on medications prescribed for control of motor symptoms, those taking such medications are more likely to have increased rates of healthcare utilization and thus may have more opportunity for prescription of an antidepressant medication or consultation with a mental health provider.

Furthermore, 15.9% of PD patients reporting depressive symptoms indicated depression, anxiety, or another emotional problem limited their physical functioning, significantly more than in the control group (Table 3). This is consistent with reports of functional impairment for PD patients with depression [9,11,16,34]. As depression is a major predictor of health-related quality of life [15] and treatment can improve perceived physical functioning [11,20], increased appropriate diagnosis and treatment of depression is one avenue to approach these limitations in physical functioning due to an emotional problem.

Although a treatment gap remains, patients report a willingness to talk to physicians about their mental health and willingness to work with a mental health professional as part of a treatment plan [35]. This concurs with physician preferences to treat depression in PD patients non-pharmacologically as a first approach [22] but is complicated by barriers to coordinating such care, including a lack of referrals, dearth of provider availability, and greater need for patient education [36,37]. While depression was considered by some patients to be an expected part of Parkinson’s, health-related quality of life can improve with recognition and treatment [35]. Depression is more closely related to patient self-perception of handicap rather than clinician assessment [12], further emphasizing the need to evaluate and treat depressive symptoms as a distinct comorbidity with well-documented influence on motor symptoms. Future work should focus on identifying variables that may contribute to receipt of depression treatment and the impact of depression treatment on physical functioning and health-related quality of life, as well as unique characteristics of this cohort compared to the general population.

This study has several limitations. The exclusion of participants’ missing responses to any primary outcome responding “I don’t know,” or declining to answer any primary outcome limits the sample size and may alter the estimates of any of the primary outcomes. Not all individuals with a formal diagnosis of Parkinson’s disease are treated with anti-parkinsonian medication; thus there may be some underreporting of PD in this study. Additionally, it is possible that depression was also misclassified in some participants as not all individuals diagnosed with depression may have screened positive for depression on the day of their Mobile Examination appointment. Anti-parkinsonian medication use was utilized as proxies for diagnoses of Parkinson’s disease, and antidepressants were assumed to be prescribed for treatment of depression. These medications may also be used for other indications [38], and there may be gaps in the medications recorded by NHANES or used as markers in this study. A visit to a mental health professional in the last 12 months was used as an indicator for non-pharmacologic treatment of depression. Such a visit, however, could have a variety of purposes, including treatment for another mental illness, assessment of mental health rather than treatment, or other counseling. Additionally, this does not necessarily reveal ongoing treatment for depression as this variable does not account for the frequency of visits within the last 12 months. The variable chosen for limitation in physical functioning includes anxiety or another emotional problem in addition to depression. While anxiety and depression are often comorbid, this is not always the case.

## 5. Conclusions

A cohort of individuals currently taking anti-parkinsonian medication was identified from nationally representative survey data and demonstrated a high prevalence of depression, above that of the general population. While treatment rates for depression are higher among the sample of those taking anti-parkinsonian medications than the general population, a treatment gap remains. Addressing this gap is vital to health-related quality of life and may potentially lessen functional impairment in this patient group.

## Figures and Tables

**Figure 1 ijerph-18-08596-f001:**
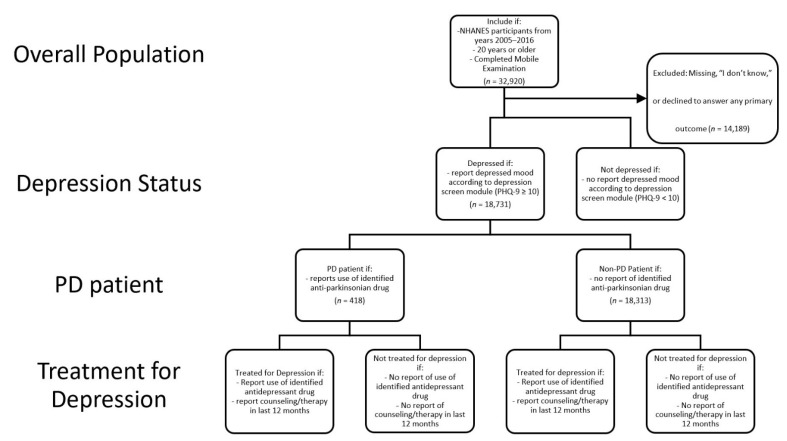
Schematic of participant sample selection, measures, and analysis.

**Figure 2 ijerph-18-08596-f002:**
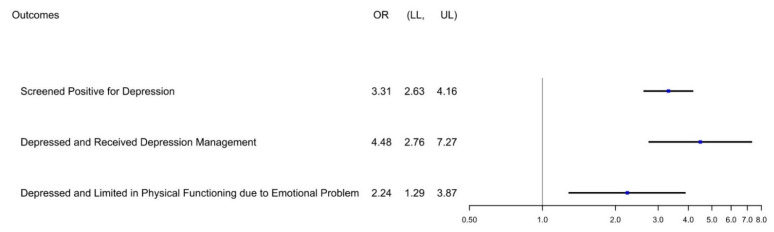
Forest Plot of Odds Ratio (OR) by PD Status. Among the entire population, OR was calculated for positive depression screen based on use of anti-parkinsonian medication. Among the portion of the population screening positive for depression (PHQ-9 ≥ 10), OR was calculated for receipt of any depression management (pharmacologic or consultation with mental health provider) and limitation of physical functioning due to depression, anxiety, or another emotional problem. LL: Lower Limit of 95% Confidence Interval. UL: Upper Limit of 95% Confidence Interval.

**Table 1 ijerph-18-08596-t001:** Study Sample Demographic Characteristics.

	Unweighted N	Weighted %
**Age**		
20–39	3514	22.64
40–59	6015	39.23
60–79	7232	30.89
80+	1970	7.24
**Gender**		
Male	8251	42.3
Female	10,480	57.7
**Ethnicity**		
Hispanic	3772	8.74
Non-Hispanic White	9464	75.34
Non-Hispanic Black	4000	10.16
Other Race	1495	5.76
**Income**		
<45,000	7182	56.18
≥45,000	9517	43.82
Missing	2032	
**Economic Status**		
Above Poverty Line	13,851	87.73
In Poverty	3365	12.27
Missing	1515	
**Education**		
Less than High School	4801	15.94
High School Graduate	9652	54.05
College Graduate	4255	30.02
Missing	23	
**Marital Status**		
Married/Living with Partner	11,009	63.92
Not Married	7712	36.08
Missing	10	
**Citizenship**		
U.S. citizen	17,243	95.65
Non-U.S. Citizen	1472	4.35
Missing	16	

**Table 2 ijerph-18-08596-t002:** Comparison of Patients With and Without Parkinson’s Disease.

	PD		No PD		*p*-Value
	(*n* = 418, 2.22%)		(*n* = 18,313, 97.78%)		
	Unweighted N	Weighted %	Unweighted N	Weighted %	
**PHQ-9 ≥ 10**					<0.0001
Yes	108	26.46	1852	9.58	
No	260	73.54	14,748	90.42	
**Age**					<0.0001
20–39	54	15.75	3460	22.8	
40–59	153	43.7	5862	39.13	
60–79	129	26.04	7103	30.99	
80+	82	14.51	1888	7.08	
**Gender**					0.0207
Male	170	34.9	8081	42.47	
Female	248	65.1	10,232	57.53	
**Ethnicity**					0.0077
Hispanic	62	6.03	3710	8.8	
Non-Hispanic White	275	81.98	9189	75.19	
Non-Hispanic Black	60	7.8	3940	10.21	
Other Race	21	4.19	1474	5.8	
**Income**					<0.0001
<45,000	105	41.19	7077	56.52	
≥45,000	260	58.81	9257	43.48	
Missing	53		1979		
**Economic Status**					<0.0001
Above Poverty Line	271	79.05	13,580	87.92	
In Poverty	107	20.95	3258	12.08	
Missing	40		1475		
**Education**					0.0004
Less than High School	129	22.67	4672	15.79	
High School Graduate	219	56.61	9433	53.99	
College Graduate	69	20.73	4186	30.23	
Missing	1		22		
**Marital Status**					0.001
Married/Living with Partner	208	54.84	10,801	64.13	
Not Married	209	45.16	7503	35.87	
Missing	1		9		
**Citizenship**					0.0017
U.S. citizen	401	97.99	16,842	95.6	
Non-U.S. Citizen	17	2.01	1455	4.4	
Missing	0		16		

**Table 3 ijerph-18-08596-t003:** Treatment of Depression in Individuals Screening Positive for Depression (PHQ-9 ≥ 10).

	Overall		PD Status				*p*-Value
			Yes		No		
			(*n* = 108, 5.75%)		(*n* = 1852, 94.25%)		
	Unweighted N	Weighted %	Unweighted N	Weighted %	Unweighted N	Weighted %	
**Depression Management by Medication or Mental Health Encounter**							<0.0001
Yes	977	54.97	87	85.12	890	53.13	
No	983	45.03	21	14.88	962	46.87	
**Depression Managed by Medication**							0.0002
Yes	723	42.51	63	62.42	660	41.3	
No	1237	57.49	45	37.58	1192	58.7	
**Limited Physical Function Due to Emotional Problem**							0.0052
Yes	163	9.18	17	15.9	146	8.77	
No	1738	90.82	86	84.1	1652	91.23	
Missing	59		5				
**Consulted Mental Health Provider in Last 12 months**							<0.0001
Yes	599	32.51	68	67.71	531	30.36	
No	1360	67.49	40	32.29	1320	69.64	
Missing	1		0				

## Data Availability

The datasets generated and/or analyzed during the current study are available in the National Health and Nutrition Examination Survey (NHANES) repository, https://www.cdc.gov/nchs/nhanes/index.htm (accessed 12 July 2019).

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
