# Peer review of "Exploring Treatment for Depression in Parkinson’s Patients: A Cross-Sectional Analysis"

_ijerph, 2021, doi:10.3390/ijerph18168596_

Round 1

Reviewer 1 Report

In this manuscript, authors concluded ”treatment gap remains". What do authors mean?  And authors maintained "addressing this gap is vital to health-related quality of life".  How do authors conclude the relationship between difference of depressive medication in two group and their OOL?

Authors should explain the relationship between depressive medication and QOL by presenting data.

Reviewer 2 Report

Dear authors,

Thank you for your work in addressing this important issue. I gather from your results that a positive screen for depression occurred more commonly in participants with identified Parkinson’s disease (26% vs. 10%) and was associated with more impairment in physical functioning (11% vs. 5%). Among participants who screened positive for depression, those with PD were more likely to be receiving mental health care (85% vs. 53%).

I appreciate that your study size was large and that the PHQ-9 was appropriately used as a screen for depression. 

I have a number of queries as listed below, among which I feel most strongly about Points 8 and 9 in the Discussion.

Abstract

  1. Lines 18-19 repeat what is already said in line 11; consider spending the word count on more details of your results instead.

Introduction

  1. Line 28 makes it sound as if 2020 hasn’t happened yet – perhaps consider rephrasing? 

Methodology

  1. What aspects of the Mobile Examination were assessed in the study? All the results appear to be based on questionnaires – if so, it is unnecessary to mention the Examination.
  2. Primary outcome (line 88) should be much more specific than “treatment for depression” – it completely leaves out the comparative aspect of your results.
  3. Were mental health care options discussed with participants who screened positive for depression but were not receiving care?

Results

  1. Please indicate some absolute numbers in the text, particularly the total number of participants, number of participants with PD and number of participants who screened positive for depression.
  2. Suggestions for your consideration: the text (lines 185-91) associated with Table 4 gives useful information. Adding odds ratios (e.g. OR of screening positive for depression in the presence vs. absence of PD) may help the reader understand a bit more clearly. Expanding the text may also enable you to do away with Table 4, which I personally find dilutes and muddles the impact of Table 3. For instance, what is the value of Table 4’s “Consulted Mental Health Provider” row? You’ve already presented this information specific to depression in Table 3. Why bring it up again in the context of the whole 19,000 participants, who could have consulted a mental health care provider for anything from schizophrenia to autism?

Discussion

  1. Your results indicate a treatment gap of 15% for depression in patients with PD. How does this compare with treatment gaps for depression in other populations? A 2004 study found a treatment gap of 56% for depression (Kohn et al, Bulletin of the WHO). Certainly 15% is not 0%, but it is pertinent to consider how bad the problem is relative to the real world. It will also help to guide how strongly to word your manuscript title and recommendations.
  2. Your results suggest that depressed patients without PD are the ones who require more attention and help. Table 3 shows they are thrice as likely to not be under care. Surely this deserves some discussion?
  3. I disagree with the phrase “positive screens among those being treated for depression may indicate that PD patients are undertreated for depression” (line 198-99). Depression takes time to treat; your analysis was cross-sectional. Furthermore, depression presents with varying severity and trajectories. I would refrain from commenting on the quality of treatment in the absence of a longitudinal assessment with consideration of illness-related factors.
  4. Line 204 feels like it should read “prevalence of depression in non-PD patients in our study…”
  5. How many participants were excluded due to answering ‘I don’t know’/ declining to answer? How may their exclusion have affected your results?

Thank you for considering these queries. All the best. 

Reviewer 3 Report

Manuscript ID:  ijerph-1297270
Type of manuscript: Article
Title: Treatment for Depression is Underutilized in Parkinson’s Patients: A
Cross-Sectional Analysis

Authors: Elisabeth C. DeMarco, Noor Al-Hammadi, Leslie Hinyard *

Review Report:

The article by Hinyard and colleagues identifies a cohort of patients with Parkinson’s disease from nationally representative survey data and analyses trends in depression care. The authors have used data from the 2005-2006 through 2015-2016 waves of the National Health and Nutrition Examination Survey (NHANES). In this analysis, individuals were classified as Parkinson’s patients by reported medication use. PHQ scores were used to identify individuals screening positive for depression. In this analysis, a composite treatment variable examined reported use of mental health services and anti-depressant medication. The authors report that survey participants reporting use of anti-parkinsonian medication screened positive for depression, reported use of anti-depressant medication, and reported visits to mental health services more frequently than the control group. A cohort of individuals with Parkinson’s disease from nationally representative survey data was identified and confirmed previous findings – depression is highly prevalent among individuals with Parkinson’s disease and this common comorbidity is often inadequately treated. However, they concluded that further research is required to determine the reason for this under treatment and necessary interventions to combat this problem. This paper lies within the aims and scope of mental health section of IJERPH journal as it highlights the high prevalence of depression, an under-recognized and under-treated comorbidity of Parkinson’s Disease. 

Overall Recommendation: Accept after Minor Revisions: The paper is in principle accepted after revision based on the reviewer’s comments.

Review Report:

However, there are few minor points that need to be addressed:

  1. Some tables lack numbering. After Table 2 there is no numbering.
  2. A schematic to explain the rational step wise analysis of the use of anti-depression medications amongst patients with PD like symptoms should be highlighted.
  3. Can a chart comprising the drugs used by Parkinson patients for depression be highlighted? Drug combination therapy (all combinations used) amongst patients be highlighted against drug medication prescribed for people with PD without depression, people without PD and people with depression and not PD.

Control patients : None of the drugs

Depression only patients : only antidepressants

PD patients without depression : only PD drugs

PD patients with depression : both PD + anti-depressants

All the drug combinations should be charted in order to get a clearer persceptive and making the points more cleared to the reader.

Round 2

Reviewer 1 Report

I have no more comment against this manuscript.

Reviewer 2 Report

Dear authors, thank you for considering the suggestions with care. I feel that the addition of participant numbers and odds ratios has made your results more comprehensive. The new discussion about treatment gaps feels more balanced and cognisant of real-world situations. 

Overall I feel your manuscript to be much improved. I only have a couple more minor queries:

  1. Including more figures has made your study process easier to grasp. However, Figures 1 and 2 seem to have some overlap. Perhaps they could be combined?
  2. I know what you mean by "some sort of treatment" but in a journal article it sounds a bit flippant; would you consider rephrasing that?

Thank you and all the best in your future endeavours.
